# S4RL: Surprisingly Simple Self-Supervision for Offline Reinforcement Learning in Robotics

**Samarth Sinha**[1,2*]**, Ajay Mandlekar**[3]**, Animesh Garg**[2,4]

[1] Facebook AI Research, [2]University of Toronto, Vector Institute, [3]Stanford University, [4]Nvidia

**Abstract:** Offline reinforcement learning proposes to learn policies from large collected datasets without interacting with the physical environment. These algorithms have made it possible to learn useful skills from data that can then be deployed in the environment in real-world settings where interactions may be costly or dangerous, such as autonomous driving or factories. However, offline agents are unable to access the environment to collect new data, and therefore are trained on a static dataset. In this paper, we study the effectiveness of performing data augmentations on the state space, and study 7 different augmentation schemes and how they behave with existing offline RL algorithms. We then combine the best data performing augmentation scheme with a state-of-the-art Q-learning technique, and improve the function approximation of the Q-networks by smoothening out the learned state-action space. We experimentally show that using this Surprisingly Simple Self-Supervision technique in RL (S4RL), we significantly improve over the current state-of-the-art algorithms on offline robot learning environments such as MetaWorld [1] and RoboSuite [2, 3], and benchmark datasets such as D4RL [4].

**Keywords:** Offline Reinforcement Learning, Data Augmentation, Self-Supervised Learning

## 1 Introduction

In reinforcement learning (RL), an agent is trained to interact with its environment and learn useful skills that help with solving the given task. However, interacting with the environment may be costly, or unsafe to do from scratch in real-world scenarios such as self-driving or industrial robotics. Unlike directly interacting with the environment, it is much cheaper to collect and store data such that an agent can be trained on it and learn from the past experience. In direct contrast to reinforcement learning, offline RL (or batch RL) [5] allows a setup where a behavioral policy is used to collect and store the data such that a target policy can then be used to train on the data, without any further interactions with the environments. The behavioral policy can range from human demonstrations, random policy in the given environment, or near-optimal policies for the task. However, learning from such demonstrations is difficult since the data does not cover the full state-action space, and naive behavioral policies will not cover the state-action distribution for an optimal policy for a task.

Current offline model-free RL algorithms include learning a $Q$-function [6] where a parameterized neural network is trained to learn the state-action values from data [7–12]. This family of algorithms suffer from overestimating the true state-action values of data that is not in the same distribution as the offline dataset used. Along with overestimation error, another source of error is with the function approximation of the neural networks that are typically used to parameterize the $Q$-functions. This form of function approximation error may be of many forms, and recent methods have employed self-supervised learning, causal inference [13] and data augmentations to the image observations during training, resulting in state-of-the-art results for normal reinforcement learning [14–16]. However, it is unclear to how perform such augmentations from proprioceptive information and the role

---

* Work primarily done when SS was at University of Toronto.

5th Conference on Robot Learning (CoRL 2021), London, UK.

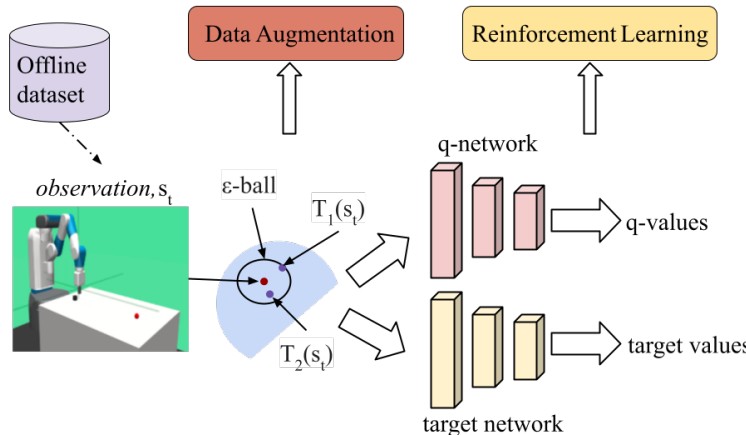

**Figure 1:** Overview of the proposed augmentation scheme. Here $T_1(s_t)$ and $T_2(s_t)$ represent different augmentation schemes. We simply perform $k-$ data augmentations to the observations $s_t$, and use that the augmented states to calculate the q-values and target values for better function approximation of the Q-network by ensuring that local perturbations in the state space have similar $Q$-value estimates.

of augmentation in offline RL. In this paper we investigate the role of data augmentations from proprioceptive observations focused on offline RL. Data augmentation also help perform better function approximation for $Q$-networks since it allows to smoothen out the state space by "visiting" the local regions and ensuring the learned $Q$-values are similar throughout, since small perturbations in the given observation should not lead to drastically different $Q$-values.

Our main contribution is a Surprisingly Simple Self-Supervised offline RL (S4RL) algorithm that combines studying data augmentations with a simple $Q$-learning method to significantly improve the performance of offline RL algorithm. The proposed framework is general and can be added to a number of off-the-shelf offline/batch RL algorithms for robot learning. We present comparisons our model to other comparable self-supervised learning strategies and state-of-the-art offline RL methods on the D4RL benchmark dataset [4], which consists of continuous control, navigation and robotic control tasks collected from suboptimal and human agents. We further experiment with two dexterous robot manipulation environments: MetaWorld [1] and RoboSuite [2, 3]. Across all evaluations, we find that our proposed state-based data augmentation for $Q$-learning significantly outperforms base offline RL algorithm as well as various competitive data augmentation baselines.

## 2    Related Work

**Offline reinforcement learning:** In offline RL, a static dataset of demonstrations is collected, and is used to train an agent for a given environment [5]. Popular actor-critic algorithms, such as Soft Actor Critic (SAC) [17], tend to perform poorly on learning from offline datasets, because they are unable to generalize to out-of-distribution (OOD) data because of an overestimation bias: where the critic overestimates the values of state-action pairs not previously encountered from the collected data which leads to a brittle policy.

The convergence properties of value functions have been proven in simple settings [18]; however learning a policy using the learned value functions remains an active challenge in offline RL. Simple algorithms like fitted $Q$-iterations have been able to learn state-action value functions, but do not work well with sub-optimal data [19, 20]. To help solve the generalization and extrapolation problem, different offline RL algorithms have proposed to use constrained policy optimization between the target policy and the behaviour policy that was used to collect the data. These constraints include using KL-divergence [10, 21], MMD [8] or $f$-divergence [22]. Some algorithms have also proposed to learn state-conditioned VAEs to minimize the *extrapolation error* [7, 8, 23]. Recent work by Buckman et al. [24] investigated using worst-case analysis to train a policy, denoted as *pessimism*. Conservative $Q$-Learning (CQL) explicitly deals with the problem of overestimation for OOD state-action distribution by *pushing down* the value of randomly sampled state-action distributions in the critic objective [25]. Learning a dynamics model from offline data has been promising [26, 27]. Recent work has also shown great promise using offline RL for real robotics tasks by learning policies

that can directly be deployed on real robots to perform complex tasks [28–30]. Similar to before, such work relies on collecting large scale robotic datasets [3, 31, 32], and learning policies that can then be safely deployed after a few steps of finetuning.

**Representation learning in RL:** Recent work in state and pixel-based RL suggests the need to learn better representations from data usig better network architectures [33] or self-supervision [14]. Self-supervised representation learning has recently been applied to learn RL agents from pixel-data. By using data augmentations and contrastive learning, Laskin et al. [14] showed significant improvements on learning from pixel data. The need for contrastive learning was simplified by RL with Augmented Data (RAD) [16] and Data-Regularized $Q$-learning (DrQ) [15] as they provide a simpler alternative that used data augmentations without a contrastive objective. More recently, Self-Predictive Representations (SPR) optionally uses data augmentations for a self-supervision [34]. The closest work to S4RL include DrQ [15] and RAD [16] in that we utilize their method for $Q$-learning over augmentations, but in contrast we focus specifically on offline RL for robotic tasks and propose augmentations from states; we also benchmark different self-supervision techniques from states. There has also been recent works showing the effectiveness of simple data augmentation in imitation learning [35, 36].

## 3  Preliminaries

In reinforcement learning, an agent interacts with an environment to learn an optimal policy. This is typically framed as a Markov Decision Process (MDP) which can be represented by a tuple of the form $(\mathcal{S}, \mathcal{A}, \rho, r, \gamma, \rho_0)$, where $\mathcal{S}$ is the state space, $\mathcal{A}$ is the action space, $\rho(s_{t+1}|s_t, a_t)$ is the transition function given the current state and action pair, $r(s_t)$ is the reward model given the state, $\gamma$ is the discount factor where $\gamma \in [0, 1)$ and $\rho_0$ is the initial state distribution of the MDP.

A policy, represented by $\pi : \mathcal{S} \to \mathcal{A}$ is trained to maximize the expected cumulative discounted reward in the MDP. This can simply be formalized as: $\mathbb{E}\left[ \sum_{t=0}^{\infty} \gamma^t r(s_t, \pi(a_t|s_t)) \right]$

Furthermore, a state-action value function, $Q(s_t, a_t)$, is the value of performing a given action $a_t$ given the state $s_t$. The $Q$-function is trained by minimizing the Bellman Error over $Q$ in a step called policy evaluation

$$Q_{i+1} \leftarrow \arg\min_Q \mathbb{E}\Big[ r_t + \gamma Q_i(s_{t+1}, a_{t+1}) - Q_i(s_t, a_t) \Big] \tag{1}$$

where $Q_i$ is the $i$-th step of policy evaluation. The policy is then trained to maximize the state-action values of performing an action $a_t$ given $s_t$ in a step called policy improvement:

$$\pi_{i+1} \leftarrow \arg\max_\pi \mathbb{E}\Big[ Q(s_t, \pi_i(a_t|s_t)) \Big] \tag{2}$$

where $\pi_i$ is the $i$-th step of policy improvement.

Unlike traditional reinforcement learning, in offline RL, the goal is not to learn an optimal policy for the MDP, but rather to learn an optimal policy given the dataset. A behaviour policy $\mu$ is used to collect a static dataset $\mathcal{D}$ which is then used to train a target policy $\pi$. The policy improvement can now be stated as

$$\pi_{i+1} \leftarrow \arg\max_\pi \mathbb{E}_{s_t \sim \mathbb{D}}\Big[ Q(s_t, \pi_i(a_t|s_t)) \Big] \tag{3}$$

where $\mathcal{D}$ is the static dataset. Since the nature of the behaviour policy $\mu$ is unknown, and can be composed of one (or more) sub-optimal policies, the RL task becomes challenging. Since offline RL algorithms tend to generalize poorly to OOD data, the nature of $\mu$ and the optimality of $\mu$ is important. As shown Fu et al. [4], typical offline RL methods perform poorly when a mixture of suboptimal policies or an untrained random policy is used to collect the dataset $\mathcal{D}$. In deep reinforcement learning, both the policy and the value functions are parameterized using neural networks and trained using gradient descent.

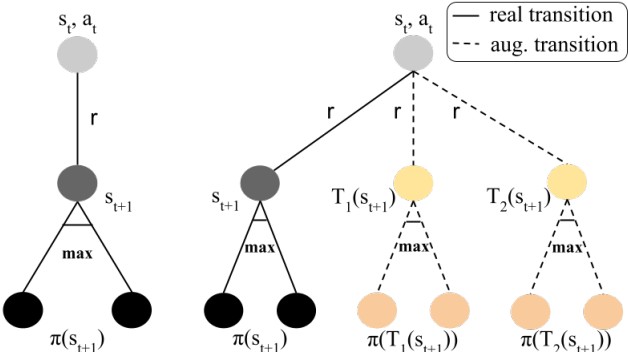

**Figure 2:** A diagram illustrating the difference between a Bellman backup for offline RL versus offline RL with augmentations. Using data augmentations, we are able smoothen out the local state-space around the observations in the dataset $s_t$.

# 4 Method

In this section we will first consider a simple $Q$-learning technique based on DrQ [15], which helps combine data augmentations and learning better value functions during training by encouraging local points around the state to have similar $Q$-values. Then, we introduce 7 different data augmentations strategies and apply them on the otherwise static dataset $\mathcal{D}$ to improve function approximation by smoothening the state-space for the $Q$-network.

## 4.1 Self-Supervision in Offline RL

We denote a data augmentation transformation as: $\mathcal{T}(\tilde{s}_t|s_t)$ where $s_t \sim \mathcal{D}$, $\tilde{s}_t$ is the augmented state, and $\mathcal{T}$ is a stochastic transformation. After producing one or multiple augmentations of a given state $\tilde{s}_t \sim \mathcal{T}(s_t)$, we need a method to incorporate the augmentation to encourage smoothness and consistency. For example, [14] uses contrastive learning to "pull" the representations of two augmentations of the same state towards each other, while "pushing" two different state representations apart from each other. We follow a similar scheme to DrQ [15], where we perform multiple augmentations of a given state, and we simple average the state-action values and target values over the different augmetations of the state.

$$\min_{Q} \mathbb{E}_{s_t, a_t \sim \mathcal{D}} \left[ r_t + \gamma \frac{1}{i} \sum_i Q(\mathcal{T}_i(\tilde{s}_{t+1}|s_{t+1}), a_{t+1}) - \frac{1}{i} \sum_i Q(\mathcal{T}_i(\tilde{s}_t|s_t), a_t) \right] \tag{4}$$

The main difference between our proposed objective and the normal $Q$-learning objective is the mean over $i$ different augmentations in the second term of the equation. We augment the Bellman error in Equation 1 to simply be the mean error over the $i$ different augmentations of the same state. Intuitively, this will help improve the consistency of the $Q$-value within some perturbation field of the current state $s_t$, since the Bellman backups are taken over $i$-different views of the same state $s_t$.

By simply averaging the state-action values and target values, we assume that the reward function is locally smooth to small perturbations in the state. Utilizing multiple augmentations further allows the networks to learn different variations of the data; we explore the role of increasing the number of different augmentations empirically in Section 5.1. We use the trained $Q$-networks to perform policy improvement using the objective in Equation 3, without augmentations. The benefits of the augmentations are distilled to the policy since the value functions are used to train the policy. By combining such local perturbations to the states with self-supervision, we are able to learn a more robust policy that can generalize better to unseen data when deployed on robots. An overview of the proposed method is available in Figure 1, where we disentangle the data augmentation and reinforcement learning steps to learn q-values and target values over $i$ different augmentations of the original state. Without loss of generality, to show local perturbations, we draw an $\epsilon$-ball over the current state $s_t$, however different versions of augmentations are possible, as will be discussed.

## 4.2 Data Augmentations

In computer vision research, augmentation are commonly used as a way to obtain the same datapoint from multiple viewpoints. Transformations such as rotation, translations, color jitters, etc. are commonly used to train neural networks. Such transformations preserve the semantics of the image after the transformation since for example: an image of a cat rotated $15°$ remains an image of a cat. However, when working from only proprioceptive information of an agent (for example: the joint angles and velocity information of an industrial robot), such transformations are semantically meaningless.

In offline RL, since an agent is unable to interact with the environment to collect new data samples, data augmentations serve as a simple technique that can allow the agent to do view different augmentations of the trajectories in the dataset. Data augmentations from states, assumes that the output of a small transformation to an input state results in a physically realizable state in the environment. If the augmented state is in the domain of the state space, $\tilde{s}_t \in \mathcal{S}$, then we consider it a valid transformation. By performing valid augmentation stratgies, we are able to artificially increase the amount of data available during training.

Even if a transformation is physically realizable in the environment, a data augmentation strategy that is too aggressive may end up hurting the RL agent since the reward for the original state may not coincide with the reward obtained from the augmented state. That is $r(s_t) \neq r(\tilde{s}_t)$. Therefore, one key assumption made when performing data augmentations on the states is that the reward model $r(s_t)$ is *smooth*, that is $r(s_t) \approx r(\tilde{s}_t)$. Therefore the choice of $\mathcal{T}(\tilde{s}_t|s_t)$ needs to be a local transformation to perturb the state without changing the semantics. The choice of $\mathcal{T}(\tilde{s}_t|s_t)$ is more important in offline RL than in traditional RL, since in traditional RL the agent may be able to self-correct for poor choices of transformations by being able to visit those states and correct the value functions using Bellman backups. A comparison is presented in Figure 2, where we see that compared to normal offline RL, adding augmentation allows for multiple paths in the policy evaluation step using Bellman backups. By using data augmentation, we seek to exploit this smoothness property of the reward model in the MDP, which enables us to artificially visit states that may be in the state space, but not in the dataset: $s_t, \tilde{s}_t \in \mathcal{S}$ but $\tilde{s}_t \notin \mathcal{D}$.

Following the constraints, we consider the following transformations $\mathcal{T}(\tilde{s}_t|s_t)$:

1. **zero-mean Gaussian noise** to the state: $\tilde{s}_t \leftarrow s_t + \epsilon$, where $\epsilon \in \mathcal{N}(0, \sigma I)$ and $\sigma$ is an important hyperparameter, which we will later discuss.

2. **zero-mean Uniform noise** to the state: $\tilde{s}_t \leftarrow s_t + \epsilon$, where $\epsilon \in \mathcal{U}(-\alpha, \alpha)$ and $\alpha$ is an important hyperparameter.

3. **random amplitude scaling** as in RAD [16]: $\tilde{s}_t \leftarrow s_t * \epsilon$, where $\epsilon \in \mathcal{U}(\alpha, \beta)$, which preserves the signs of the states.

4. **dimension dropout** which simply zeros one random dimension in the state: $\tilde{s}_t \leftarrow s_t \cdot \mathbf{1}$ where $\mathbf{1}$ is a vector of 1s with one 0 randomly sampled from a Bernoulli distribution. This transformation preserves all-but-one intrinsic values of the current state.

5. **state-switch** where we flip the value of 2 randomly selected dimensions in the state. This naive transformation will likely break the "physical realizability" assumption, since it is possible that the two randomly selected samples are semantically dissimilar properties of a robot (such as joint angle and joint velocity). To overcome this we hardcode the pairs of dimensions that can be switched for each environment (details in Appendix F).

6. **state mix-up** where we use mixup [37] between the current and the next state: $\tilde{s}_t \leftarrow \lambda s_t + (1 - \lambda)s_{t+1}$, where $\lambda \sim \texttt{Beta}(\alpha, \alpha)$ with $\alpha = 0.4$ as in [37]. The next state $s_{t+1}$ remains unchanged.

7. **adversarial state training** which we take the gradient with respect to the value function to obtain a new state. This can be written as $\tilde{s}_t \leftarrow s_t + \epsilon \nabla_{s_t} \mathbb{J}_Q(\pi(s_t))$, where $\mathbb{J}_Q$ is simply the policy evaluation update as in Equation 2, and $\epsilon$ constraints the size of the gradient. The intuition behind this is to choose the direction within an $\epsilon$-ball of $s_t$ where the $Q$-value deviates the most. This augmentation is intuitively the most useful in state spaces with high dimensionality where random noise may be less effective [38].

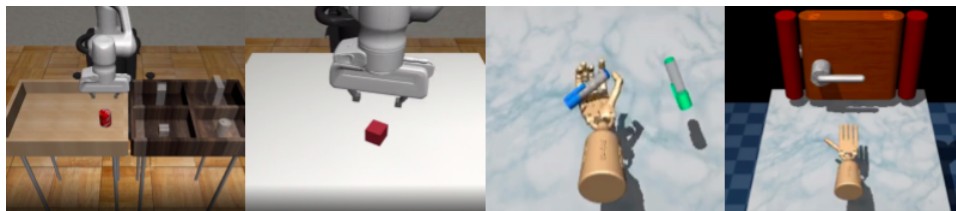

**Figure 3:** Illustration of the robotic environments used in the experiments. The environments range from robotic arms for difficult manipulation tasks such as opening a door or placing a can into a bin or dexterous manipulation of a pen to a desired configuration using a robot hand.

## 5 Experiments

In this section, we will first describe the dataset (D4RL [4]), and some hyperparameters and information about the experimental setup. Then we will use S4RL to investigate the effect of different data augmentation strategies on the OpenAI Gym subset of D4RL tasks to investigate which augmentations are useful for learning. Using the best data augmentation techniques found, we compare the algorithm to other self-supervision techniques that have been proposed for pixel-based RL, namely Contrastive Unsupervised Reinforcement Learning (CURL) [14], and SAC+AutoEncoder (SAC+AE) where we feed the augmentations to the AutoEncoder to learn generalizable representations [39]. We add S4RL framework to two state-of-the-art offline RL algorithms: Conservative $Q$-learning (CQL) [25] and Behaviour Regularized Actor Critic (BRAC) [10] to show the effectiveness over both baseline agents to show the generality of S4RL. We add the CQL regularization to each self-supervised baseline considered, since the self-supervised baselines were considered for online RL and CQL adds a regularization term to help with overestimation. We first test each baseline on the full suite of D4RL tasks [4], which includes environments such as a Maze environment where an ant agent must learn locomotion and navigation skills to reach an environment goal which requires hierarchical control, Adroit domain which requires learning dextrous object manipulation for fine-grained control of a robotic hand [40], Kitchen environment which simulates a Franka Panda robot to perform kitchen tasks. We additionally experiment with more difficult robotic environments: MetaWorld [1] which uses a simulated Sawyer robot and RoboSuite [2] which uses a Franka Panda robotic arm for manipulation. We further benchmark each of the baselines on *learning with limited data* in Appendix A, to investigate the effect of self-supervision when the agent has significantly less data to train from. Finally, we experiment with different ways to regularize $Q$-networks typically used in neural network training such as Dropout and Weight Decay in Appendix D and include ablation experiments investigating the number of augmentations and the role of hyperparameters in Appendix E.

**Hyperparameters:** For all experiments, we do not perform any hyperparamter tuning to the base CQL agent; all agents are trained using the original hyperparameters. Similar to DrQ, we use 2 augmentations for obtaining the state-action values and the target values. We use a $\sigma$ and $\alpha$ value of $3 \times 10^{-4}$ for the zero-mean Gaussian and Uniform noise augmentation variant, respectively. We use a value of $1 \times 10^{-4}$ for $\epsilon$ for adversarial training; an ablation on $\sigma$ and $\epsilon$ is available in Appendix E.

### 5.1 Results

**Role of Data Augmentations** We first investigate different data augmentation schemes and its performance on the OpenAI Gym subset of the D4RL tasks. We compare a base CQL agent with and without different forms of augmentation strategies, and tabulate the mean normalized performance, average normalized performance over all the tasks and relative rankings in Table 1. We see that using zero-mean Gaussian noise ($\mathcal{N}$), zero-mean Uniform noise ($\mathcal{U}$), state mix-up and adversarial state training consistently outperform the the baseline CQL agent, as well as different data augmentation variants. The average-ranking of S4RL-Adv and S4RL-$\mathcal{N}$ are 1.83 and 1.92 respectively, suggesting their effectiveness over a wide range of task and data distributions. We also see that CQL+S4RL agent is able to learn useful policies given data collected from a random policy as evidenced by the performance in "walker-random" where the base CQL agent is unable to learn any meaningful policy.

We also see that the S4RL agent is able to significantly outperform the baseline CQL agent on complex data distributions such as "-medium-replay" where the data collected is from all the data

**Table 1** (CQL)

| Task Name | Normal | +S4RL ($\mathcal{N}$) | +S4RL ($\mathcal{U}$) | +S4RL (Amp-Scale) | +S4RL (Dim-Drop) | +S4RL (State-Switch) | +S4RL (MixUp) | +S4RL (Adv) |
|---|---|---|---|---|---|---|---|---|
| cheetah-random | 35.4 | **52.3** | 50.3 | 46.4 | 45.5 | 40.3 | 45.2 | **53.9** |
| cheetah-medium | 44.4 | **48.8** | 47.0 | 42.5 | 45.6 | 41.2 | 46.2 | **48.6** |
| cheetah-medium-replay | 42.0 | **51.4** | 50.8 | 43.1 | 49.6 | 41.0 | 46.2 | **51.7** |
| cheetah-medium-expert | 62.4 | **79.0** | **78.5** | 71.2 | 66.1 | 68.3 | 73.1 | **78.1** |
| hopper-random | **10.8** | **10.8** | 10.7 | **10.8** | 9.3 | 9.5 | **11.0** | 10.7 |
| hopper-medium | 58.0 | 78.9 | **80.6** | 60.3 | 45.3 | 54.6 | **79.2** | **81.3** |
| hopper-medium-replay | 29.5 | **35.4** | 35.2 | 28.6 | 20.4 | 20.9 | **35.6** | **36.8** |
| hopper-medium-expert | 111.0 | **115.2** | 112.3 | 102.6 | 98.3 | 108.6 | **113.5** | **117.9** |
| walker-random | 7.0 | **24.9** | 20.5 | 6.9 | 3.2 | 9.9 | 10.2 | **25.1** |
| walker-medium | 79.2 | **93.6** | **93.9** | 81.6 | 65.3 | 65.3 | 88.6 | **93.1** |
| walker-medium-replay | 21.1 | **30.3** | **31.9** | 25.1 | 8.5 | 9.3 | 27.6 | **35.0** |
| walker-medium-expert | 98.7 | **112.2** | 104.1 | 100.2 | 86.8 | 103.8 | **104.3** | **107.1** |
| **average-score** | 49.96 | **60.57** | **59.65** | 51.60 | 45.31 | 47.73 | 56.73 | **61.6** |
| **average-ranking** | 5.83 | **1.92** | **3.00** | 5.00 | 6.42 | 6.25 | 3.17 | **1.83** |

**Table 1:** Investigating the importance of **different augmentation schemes in offline RL**. We train a baseline CQL agent, and self-supervised CQL+S4RL agents using different augmentation schemes, and report the **mean normalized scores over 5 random seeds**. We report the baseline CQL results directly from Fu et al. [4]. To make the table easier to read we color the best algorithm on a task in blue, the second best in brown, and the third best in green. We also report the average-score and average-ranking found using that augmentation scheme.

**Table 2**

| Domain | Task Name | CQL Normal | +CURL ($\mathcal{N}$) | +VAE ($\mathcal{N}$) | +S4RL ($\mathcal{N}$) | +S4RL (Mix-Up) | +S4RL (Adv) | BRAC-v Normal | +S4RL ($\mathcal{N}$) | +S4RL (Mix-Up) | +S4RL (Adv) |
|---|---|---|---|---|---|---|---|---|---|---|---|
| AntMaze | antmaze-umaze | 74.0 | 70.3 | 86.1 | **91.3** | 86.7 | **94.1** | 70.0 | 76.5 | 66.4 | **80.3** |
| | antmaze-umaze-diverse | 84.0 | 81.4 | 85.1 | **87.8** | 86.9 | **88.0** | 70.0 | 81.3 | 65.4 | **80.9** |
| | antmaze-medium-play | 61.2 | 60.9 | 62.1 | 61.9 | 61.3 | 61.6 | 0.0 | 0.0 | 0.0 | 0.0 |
| | antmaze-medium-diverse | 53.7 | 45.4 | 59.9 | 78.1 | 62.3 | 82.3 | 0.0 | 0.0 | 0.0 | 0.0 |
| | antmaze-large-play | 15.8 | 12.3 | 15.2 | **24.4** | 16.7 | **25.1** | 0.0 | 0.0 | 0.0 | 0.0 |
| | antmaze-large-diverse | 14.9 | 8.3 | 17.3 | **27.0** | 15.9 | **26.2** | 0.0 | 0.0 | 0.0 | 0.0 |
| Gym | cheetah-random | 35.4 | 43.1 | 35.7 | **52.3** | 45.2 | **53.9** | 31.2 | **35.6** | 34.3 | **36.1** |
| | cheetah-medium | 44.4 | 44.8 | 45.6 | **48.8** | 46.2 | 48.6 | 46.3 | **49.1** | 46.1 | 46.0 |
| | cheetah-medium-replay | 42.0 | 36.5 | 41.9 | 51.4 | 46.2 | **51.7** | 47.7 | **47.9** | 47.0 | 47.9 |
| | cheetah-medium-expert | 62.4 | 65.6 | 70.7 | **79.0** | 73.1 | 78.1 | 41.9 | **52.1** | 46.7 | 53.4 |
| | hopper-random | **10.8** | **10.8** | **10.8** | **10.8** | **11.0** | 10.7 | 12.2 | **12.9** | 10.9 | 12.1 |
| | hopper-medium | 58.0 | 61.9 | 66.4 | 78.9 | 79.2 | 81.3 | 31.1 | 45.1 | 41.9 | 48.0 |
| | hopper-medium-replay | 29.5 | 30.1 | **34.8** | 35.4 | 35.6 | 36.8 | 0.6 | 1.1 | 0.9 | 0.4 |
| | hopper-medium-expert | **111.0** | 109.7 | 114.3 | 113.5 | 112.3 | 117.9 | 0.8 | 59.0 | 49.5 | 55.3 |
| | walker-random | 7.0 | 6.6 | 6.5 | **24.9** | 10.2 | **25.1** | 1.9 | 9.5 | 2.4 | 14.1 |
| | walker-medium | 79.2 | 81.3 | 80.8 | **93.6** | 88.6 | **93.1** | 81.1 | **85.4** | 80.9 | 86.7 |
| | walker-medium-replay | 21.1 | 24.5 | 26.4 | 30.3 | 27.6 | **35.0** | 0.9 | 1.1 | 0.9 | 1.1 |
| | walker-medium-expert | 98.7 | 103.0 | 99.6 | **112.2** | 104.3 | 107.1 | 81.6 | 90.4 | 89.7 | 94.5 |
| Adroit | pen-human | 37.5 | 33.4 | 39.7 | **44.4** | 41.6 | **51.2** | 0.6 | **11.3** | 0.8 | **14.1** |
| | pen-cloned | 39.2 | 40.1 | 41.3 | **57.1** | 44.5 | **58.2** | -2.5 | **10.6** | 0.5 | 9.9 |
| | hammer-human | 4.4 | **6.1** | **6.0** | 5.9 | 6.3 | 6.3 | 0.2 | **3.2** | 3.3 | 5.4 |
| | hammer-cloned | 2.1 | 1.9 | 2.1 | **2.7** | 2.4 | 2.9 | 0.3 | 0.1 | 0.3 | 1.2 |
| | door-human | 9.9 | 10.3 | 14.3 | 27.0 | 16.7 | 35.3 | -0.3 | **2.8** | 0.2 | 4.2 |
| | door-cloned | 0.4 | 0.4 | 0.7 | **2.1** | 0.3 | 0.8 | -0.1 | 0.3 | -0.1 | 0.5 |
| | relocate-human | **0.2** | **0.2** | **0.2** | **0.2** | **0.2** | **0.2** | -0.3 | 0.1 | 0.1 | 0.2 |
| | relocate-cloned | **-0.1** | **-0.1** | -0.3 | **-0.1** | **-0.1** | **-0.1** | -0.3 | -0.1 | -0.1 | -0.1 |
| Franka | kitchen-complete | 43.8 | 54.4 | 42.1 | 77.1 | 60.4 | **88.1** | 0.0 | 0.0 | 0.0 | **12.3** |
| | kitchen-partial | 49.8 | 59.6 | 49.9 | 74.8 | 59.0 | **83.6** | 0.0 | 0.0 | 0.0 | **13.9** |

**Table 2: Full set of experiments on the D4RL suite of tasks.** We perform experiments over 4 different domains and many different environments and data distributions for each environment. We compare two S4RL variants ($\mathcal{N}$ and Mix-Up) with two different popular unsupervised representation learning methods and the base CQL offline RL agent. We train the CURL and VAE variants with the Gaussian additive noise variant that was empirically the most effective. We report the mean normalized performance over 5 random seeds.

collected while training a policy in the environment. Therefore the data split consists of data that ranges from a completely untrained policy (random policy), to a "medium" trained policy.

It is also important to look at the different augmentation techniques that do not help policy learning. Techniques that hurt the performance and perform worse than the baseline CQL agent include Dimension-Dropout and State-Switch. Both techniques are inspired by popular computer vision data augmentation algorithms, mainly MixMatch [41] and CutMix [42]. Since both techniques perform element-wise operations, it is likely that they omit important information about the state of the robot such as joint velocity. Without such information, it is possible that the value function is unable to reason about the environment which results in poor value estimates.

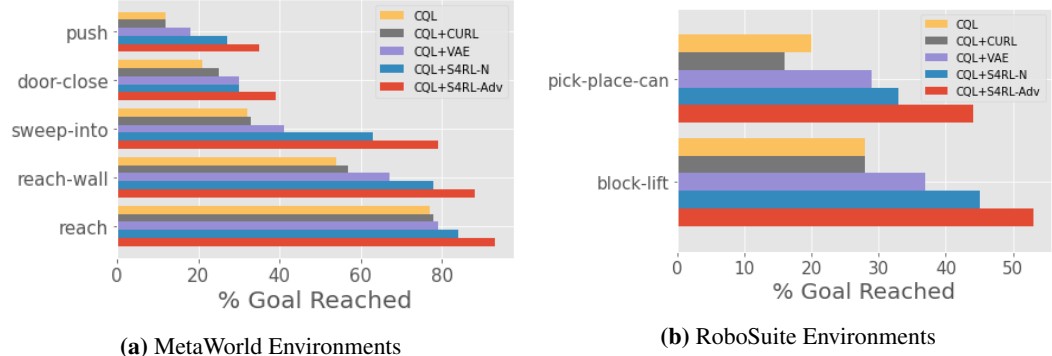

| (a) MetaWorld Environments | (b) RoboSuite Environments |

**Figure 4:** Results on Dexterous Manipulation Environments from MetaWorld [1] and RoboSuite [2, 3]. We report the fraction (as %) of goals reached in each of the environments during evaluation. Notably with a fixed amount of data, S4RL augmentation improves policy success rate over base CQL by approx. 20% across different tasks.

**Comparison to other self-supervision techniques** Following insights from Table 1, we choose three data augmentation approaches, namely S4RL-$\mathcal{N}$, S4RL-MixUp and S4RL-Adv, to compare to other self-supervision techniques. We present the results over all the different tasks in D4RL in Table 2 using base CQL [25] and BRAC agents [10]. Despite its simplicity, we continuously observe that +S4RL-$\mathcal{N}$ and +S4RL-adv agents are able to significantly outperform the baselines on almost all tasks, and is comparable to the best in the others for both offline RL algorithms. We specifically note that +S4RL-Adv significantly outperforms all other baselines when the state-dimensionality is high, such as the dexterous manipulation environment of Adroit and on Franka robot. In challenging environments that require hierarchical control such as the AntMaze environments, we see that the S4RL agent continues to be the best performing agent. Interestingly, on certain environments namely "pen-cloned", "pen-human", "hammer-human" and "door-human" the BRAC+S4RL agent is able to learn some useful skills, whereas the base CQL agent performs poorly. This further highlights the usefulness and generality of performing state-based augmentations for offline reinforcement learning.

**Dexterous Manipulation Environments** Finally, we perform more experiments in the robotic domain using the MetaWorld [1] and RoboSuite environments on difficult manipulation tasks. For MetaWorld, we train a SAC agent [17] for 1M steps and collect 1000 trajectories of 200 episode length. For RoboSuite, we collect data at 3 different instances during training to make the data distribution more complex. More details on the data collection is available in Appendix I. The results are presented in Figure 4. Similar to before we see significant improvements over the baseline in each of the 7 environments over both domains. The proposed S4RL-Adv agent is the only agent that is able to outperform the trained behaviour policy on 3 of the 5 MetaWorld tasks, while continuing to significantly outperform the baselines by as much as 3 times, as in the case of "push-v1" and "pick-and-place-can". Full numerical results can be found in Appendix C

## 6   Conclusion

In this paper, we present S4RL: a Surprisingly Simple Self-Supervised offline RL method that uses data augmentations to improve the function approximation for $Q$-learning algorithms in offline RL. S4RL offers simplicity and ease of implementation, and can be added to any offline agent that requries $Q$-learning. We first compare and benchmark the effectiveness of 7 different data augmentation strategies from states in offline RL and then use the insights to compare against different self-supervised representation learning algorithms that have been proposed for pixel-based online RL. We observe significant performance gains over the benchmark D4RL dataset [4] and on more dexterous robot manipulation environments. We observe significant performance gains which make offline policy learning increasingly more competitive and help robot learning from only past data. Interesting future extension of this work can seek to use the suggested augmentation schemes to build better self-supervised learning algorithms.

## Acknowledgments

Animesh Garg is supported by CIFAR AI Chair, NSERC Discovery Grant RGPIN-2021-04392 and DGECR-2021-00368, University of Toronto Engineering Xseed Grants. Ajay Mandlekar acknowledges the support of the Department of Defense (DoD) through the NDSEG program. The authors would also thank Vector Institute for computing support. The authors would also like to thank Aviral Kumar for his help setting up the CQL experiments and for helpful discussions regarding the paper.

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
