# OpenReview forum: "S4RL: Surprisingly Simple Self-Supervision for Offline Reinforcement Learning in Robotics"
_robot-learning.org/CoRL/2021/Conference — CoRL2021 Poster_

### Official Review · Reviewer_TxK8 · 2021-07-07

**Originality:** Good
**Technical Quality:** Very Good
**Clarity Of Presentation:** Very Good
**Impact:** 3

**Recommendation:**

Weak Accept: I recommend accepting the paper, but will not argue for my recommendation if the majority of other reviewers have a different opinion.

**Summary:**

This paper addresses the problem of improving the performance of
offline reinforcement learning (RL) through data augmentation. In
particular, the authors discuss ways of adding random perturbations to
the dataset in a manner that preserves the physical meaning of the
data (e.g., realistic action-reward pairs). The authors discuss a
number of data augmentation methods and present extensive evaluation
over multiple datasets.

**Issues:**

N/A

**Reviewer Expertise:**

Very good: Comprehensive knowledge of the area

**Strengths And Weaknesses:**

Offline RL is an important problem, especially in sim-to-real tasks
where performing online RL might be difficult and time-consuming. As
such, addressing the generalization issues with offline RL is
certainly an interesting and important research direction.

The paper is well written, and the problem is clear. The idea of using
data augmentation is a nice and simple way to improve the performance
of modern classifiers, especially in settings where training data is
not overly abundant.

The results look promising. It would be interesting to see if the
proposed method can be used on benchmarks where the observations are
images or other high-dimensional sensor data. Performing physically
consistent data augmentation in such settings would be more
challenging but potentially more useful as well.

One comment I have is that authors mention in a number of places in
the paper that the proposed approach reduces overfitting, yet this
claim is not properly evaluated. Overfitting is indeed a major concern
in (offline) RL since the learned policy is very much dependent on the
training set (presumably collected using other predefined
policies). One way to evaluate generalization error would be to add
process/measurement noise to the various RL environments and perform a
comparison in that setting as well.

**Summary Of Recommendation:**

This paper tackles a challenging problem, and the results look promising. The proposed data augmentation method makes sense, as it has been applied successfully in supervised learning settings.

---

> ### Author Response · Authors · 2021-08-24
> **Thank you for your time!**
>
> Thank you for your time to review our draft! We address each point individually.
>
> > Experiments with images for offline RL
>
>
> Thank you for the suggestion. The primary scope of the paper primarily revolved around state-based information for the agent.
> Other papers have considered structured pixel-based augmentations such as changing the parameters of the rendering process, such as light, texture, crops. This structure has resulted in positive effects in robustness of learning in RL [[1](https://arxiv.org/abs/2004.04136),[2](https://arxiv.org/abs/2004.14990)].
> Owing to the structural differences in underlying representations, it is not clear if which of the perturbation schemas will work. However the experiments below show the effects of measurement noise where the measurement functions maps from states to states rather than pixels to states.
> However a thorough investigation such as ours for pixel based perturbations, especially randomized and adversarial image based perturbations remains an avenue for future work.
>
>
>
> > Experiments with adding process / measurement noise to the real environment
>
>
> Thank you for the suggestion, we have added additional experiments as suggested to measure the ability of the learned offline agents to deal with measurement noise.
> During testing, we randomly perturb the states by using an additive Gaussian noise of the form N(0, 0.001).
> We report results on the same Metaworld tasks as in Figure 4 (left) and Table 3. In the results we see that we are able to significantly improve the generalization ability of the learned agents using S4RL.:
>
> |            | CQL | CQL+S4RL-N | CQL+S4RL-Adv |
> |------------|-----|------------|--------------|
> | push       | 49% | 69%        | **73%**      |
> | door-close | 38% | 65%        | **67%**      |
> | sweep-into | 21% | 50%        | **59%**      |
> | reach-wall | 15% | 19%        | **28%**      |
> | reach      | 10% | 13%        | **22%**      |

---

> > ### Comment · Reviewer_TxK8 · 2021-08-28
> > **Response**
> >
> > I would like to thank the authors for addressing the reviewers' comments. The new experiment does indicate that the proposed method improves overfitting over the baseline, so my concerns have been addressed.

---

> > > ### Author Response · Authors · 2021-08-28
> > > **Thank you for your time!**
> > >
> > > Thank you for your time and for checking the rebuttal! If you have any suggestions for what will make you consider increasing your score, then we will be happy to answer questions or add additional experiments.

---

> ### Comment · Reviewer_TxK8 · 2021-08-31
> **Post-Rebuttal Comment**
>
> I appreciate the authors' efforts during the rebuttal stage. The additional
> experiments suggest that the proposed method does improve overfitting, so
> I will keep my recommendation as a weak accept.

---

> > ### Comment · Reviewer_oh9V · 2021-09-01
> > **Reviewer-to-reviewer comment: the authors dropped all mentions of overfitting**
> >
> > Hi Reviewer TxK8, please note that the authors ended up agreeing with me that none of their experiments support the claim that overfitting has been reduced.  They have deleted all mentions of overfitting from their draft.  I wonder what you think of that development?
> >
> > Also to clarify: your suggested experiment of adding measurement noise is interesting, but any analysis of overfitting in that setting would be conflated by the closed-loop stability of the system.  I gave the authors a different suggestion for how to analyze overfitting directly but they chose to delete its mention instead.
> >
> > Thoughts?

---

> > > ### Author Response · Authors · 2021-09-01
> > > **Regarding overfitting**
> > >
> > > Thank you for your time.
> > >
> > > >  Also to clarify: your suggested experiment of adding measurement noise is interesting, but any analysis of overfitting in that setting would be conflated by the closed-loop stability of the system.
> > >
> > > The experiment with measurement noise does show that the learned agents policies that are more robust than the baselines.
> > >
> > > However, the problem of under fitting and overfitting in deep RL is somewhat ill understood as recent papers in self supervised learning have argued that they minimize the overfitting problem for Q-networks while other papers (see [[1]](https://arxiv.org/abs/2010.14498)) argue that underprameterization (leading to underfitting) may also be an issue. To avoid making contentious claims, we have softened the stance of the paper to avoid the overfitting argument, as there is no consensus, which is also in accordance to your prior suggestions.
> > >
> > > While decided to dial down the claims but we respectfully disagree with your technical argument or your interpretation of overfitting in this context. We changed the text mainly because it could be contentious since as this is an open question in the community. We plan to include  the network size experiment in the appendix in reference to overfitting without a judgmental opinion about overfitting and its role, but instead to add recorded experimental evidence of the phenomenon.
> > >
> > > >  I gave the authors a different suggestion for how to analyze overfitting directly but they chose to delete its mention instead.
> > >
> > > The experiment suggested by the reviewer regarding toy gridworld environments to compute Q* is insufficient as it does not scale to real world robotics tasks and making strong conclusion does not provide support for the argument. And the tabular gridworld domain experiment will add unnecessary jargon for the sake of technical complexity without adding real insight if this could work in a problem of realistic size, as in the case of robot learning environments.

---

> > > > ### Comment · Reviewer_TxK8 · 2021-09-01
> > > > **Overfitting, cont'd**
> > > >
> > > > Dear authors,
> > > >
> > > > Thank you for the clarification. Since the rebuttal phase is over, we will continue our discussion internally, just to be fair to other authors.

---

### Official Review · Reviewer_oh9V · 2021-07-21

**Originality:** Fair
**Technical Quality:** Poor
**Clarity Of Presentation:** Fair
**Impact:** 4

**Recommendation:**

Weak Accept: I recommend accepting the paper, but will not argue for my recommendation if the majority of other reviewers have a different opinion.

**Summary:**

This paper studies the effect of data augmentation on offline RL algorithms.  Essentially it applies the DrQ data augmentation technique to an offline-RL, proprioceptive-only-state setting.  Specifically, it measures the effect of up to 7 different data augmentations, across 3 different simulated task domains (D4RL, MetaWorld, and RoboSuite), and to two different offline-RL algorithms: CQL and BRAC.  The primary empirical results are that the noisy data augmentation can improve closed-loop evaluations of the policies, with the two most effective methods being either adding noise from (i) a zero-mean gaussian or (ii) adversarially using the Q-function gradients.

**Issues:**

See Strengths and Weaknesses, I put a list in the *(iii) Objective critiques and room for improvement in revision.*

**Reviewer Expertise:**

Excellent: Expert knowledge on the topic of the paper

**Strengths And Weaknesses:**

To best organize my feedback and analysis, I’ve created three subsections: (i) a summary of strengths, weaknesses, and what I learned, (ii) a list of objective strengths, and (iii) a list objective critiques and room for improvement.

(i) *Summary of strengths, weaknesses, and what I learned*:

My summary of the strengths and weaknesses are that there seems to be a main strength, and several main weaknesses.  In terms of strengths, the paper’s main strength is that the quantitative improvement shown by the proposed method can be large.  This includes all three of the MetaWorld, RoboSuite, and D4RL results.  In terms of weaknesses, there are several: significant claims are not supported by data, in some cases the data supports the opposite conclusions arrived at by the authors, the scope of novelty is limited, and the consideration of real-world robotics impact isn’t considered.  My subjective recommendation for the paper will be discussed in the Summary of Recommendation, but in the next two subsections, I will try to keep the analysis objective.

I would also like to emphasize to the authors that although I’ll enumerate a large list of critiques, they are on to some interesting results, and I applaud their work so far!  Their work has shown the high potential that simple data augmentation can have on making the most of offline robotics data.  In particular their use of the adversarial augmentation is a nice connection between a couple different works, and opens interesting questions for further work. I hope my feedback encourages them to revise their work further, and I do think this line of work could have an impact on the field, even if it needs significant revision first.

(ii) *Objective strengths*

1. Within the scope of the results reported within this paper, the direct comparisons show that the data augmentation methods (particularly Gaussian noise and adversarial training) improve the closed-loop evaluation results of both CQL and BRAC on the evaluated simulated tasks.
2. Comparing as well against numbers from other papers, it seems that for many of the D4RL tasks these “CQL+S4RL” numbers would be considered state-of-the-art quantitative numbers on D4RL which is a commonly used offline RL simulated benchmark.

(ii) *Objective critiques and room for improvement in revision*

1. A central claim of the paper is that the presented method reduces overfitting, but there is insufficient data to support this claim.   “Preventing overfitting” is a specific thing, which means that the model fits the training data much better than the test data. This was not measured in the presented experiments.  No analysis of fitting the training data was presented, so overfitting can’t be established.  Since the evaluations all use the metric of closed-loop evaluations, the performance increases that are seen could actually be from a phenomenon besides overfitting, for example having better closed-loop stability.  The models with the data augmentations could even overfit *more*, but have better closed-loop stability.  The authors could distinguish these effects with more experimentation.  The results of such experiments might end up changing their story considerably.
2. To clarify, this overfitting claim is a central claim of the paper that is used both to describe a limitation of prior work  (in abstract: “However, current algorithms overfit to the dataset they are trained on”; in Introduction “neural networks may overfit the data, further resulting in poor generalization…”) and as the key effect of the presented method (end of intro Method paragraph, Section 4: “Then we introduce 7 data augmentations… to reduce overfitting and improve function approximation by smoothening the state-space.”; second paragraph Section 4.2: “by performing valid augmentation… thereby reducing chances of the overfitting to the data”.)
3. The data potentially supports the opposite of the overfitting claims: Appendix C shows that Dropout makes the results worse.  If Dropout, which is well known to prevent overfitting, makes the results worse, this may support the conclusion that the phenomenon does not have to do with overfitting.  This is not discussed.  Instead, the reader is left to assume that the presented data augmentation strategy can reduce overfitting, but Dropout can not.
4. A key statement about novelty is incorrect, and reduces the scope of the novelty claim: “however, it is unclear to how perform [sic] such augmentation from proprioceptive information”. This statement is incorrect because one of the cited references, specifically [15], “Reinforcement Learning with Augmented Data”, does show both methods and results for proprioceptive inputs: see the end of their Section 4 “Extension to state-based inputs” and the results on OpenAI Gym benchmark.
5. Some missing related work discussion would be around noise/augmentation in imitation methods, including http://proceedings.mlr.press/v78/laskey17a/laskey17a.pdf (online adding noise to the demonstrator actions) and https://arxiv.org/pdf/1909.06933.pdf, (Section IVC, offline adding gaussian noise to state).
6. More on novelty claims, the paper says “The main difference between our proposed objective and the normal Q-learning objective is the mean over i different augmentations in the second term of the equation.”  But this makes it seem as though averaging Bellman updates over various augmentations is a contribution of the paper.  This was a contribution of DrQ, see for example Equation 1 and/or Algorithm 1 in [14].
7. “Self-supervision”, which is a central word in the title and in the manuscript overall, is a misnomer and is not an accurate characterization. The presented method is a data augmentation method, not a self-supervision method.  This distinction is understood in the prior works, including both [14] “Image augmentation is all you need” and [15] “Reinforcement learning with augmented data”, which both accurately characterize their methods as data augmentation methods.  This is in contrast with [13] “Contrastive Unsupervised Representations for Reinforcement Learning”, which can be called an unsupervised/self-supervised method, since the formulation involves an actual self-supervised representation.  There is a relationship between data augmentation and many self-supervision methods (i.e. many of the self-supervised methods in computer vision such as SimCLR use data augmentation as a key component), but those other methods have the goal of recovering some particular self-supervised representation.  Instead, the presented paper is strictly a data augmentation method applied to the supervised offline RL objective, with no self-supervised representations used.
8. There are also various incorrect statements discussing the central topic of the paper, which is data augmentation and its effect in the studied setting:
    - The end of the second-to-last paragraph in the Introduction states: “since small perturbations in the given observation should not lead to drastically different Q-values.”  This is not in general a true statement.  It is possible for world dynamics to be sharp or discontinuous such that slightly different states of the world may lead to vastly different Q-values for an optimal policy.  What the authors could say instead is: “Our approach makes the assumption that small perturbations in the given observation do not lead to drastically different Q-values.  While this is not generally true, our results show..."
    - The start of Section 4.2 states, “In computer vision research, augmentations are commonly used as a way to obtain the same datapoint from multiple viewpoints.”  This is not true… obtaining the same datapoint from multiple viewpoints might be called some type of data fusion, but not augmentation?
    - Also near the start of Section 4.2: “Such transformations preserve the semantics of the image after the transformation.” This is not generally true, and depends on which semantics are being discussed.  For example one can apply a left/right flip (mirror) augmentation often when doing classification computer vision, but one would not usually do this for training a robot policy that for example might want to learn to put object A to the left of object B.
    - Also near the start of Section 4.2: “However when working from only proprioceptive information of an agent…, such transformations are semantically meaningless.”  Since semantics are a human-defined concept up to interpretation, one cannot generally say that transformations do not change semantics.  In fact, the authors agree with this, two paragraphs later.
    - Second paragraph of Section 4.2: “In offline RL, data augmentations serve as a simple technique that can allow the agent to do local exploration from the trajectories in the dataset.”  Although we can interpret the augmentations as something similar to imagining local explorations, it’s not true to say that the agent actually does local exploration.  Exploration would imply receiving new observations/rewards from the environment.  The authors have more accurate language to describe this phenomenon later, saying “...enables us to artificially visit states that may be in the state space.”
    - Third paragraph of Section 4.2: “Therefore the choice of \mathcal{T}() needs to be a local transformation to perturb the state without changing the semantics.”  Two paragraphs ago it was stated that transformations are semantically meaningless, but now it says that only small transformations avoid changing the semantics.  This is inconsistent.  Also, it may be helpful in revision for the authors to define their definition of semantics, or perhaps consider removing the phrase and substituting something else that is defined.
9. Although the presented method shows increased performance on the simulated benchmarks, it is not discussed whether the discussed data augmentations would in practice improve real-world robotics results.  As shown in Table 5 in Appendix D, the preferred Gaussian noise is on the order of 1e-4.  The scale of this value is not discussed, but in general the D4RL environments have observations and actions that are normalized to [-1,1] range.  So 1e-4 is 1/20,000th of the observation range.  This might be on the order of natural noise observed by real-world robot sensors, for example the accuracy of a cartesian end-effector state estimate estimated by forward kinematics from 6 noisy encoders.  Point being that perhaps natural noise on real-world robots already provides the benefits observed by the method in simulated scenarios.  This question might be investigated further in simulation, through simulating proprioceptive sensor noise.

**Summary Of Recommendation:**

I think it is clear that the paper needs considerable revision, but the question is whether that amount of revision is small enough such that it could be accepted to this conference, or if it is considerably large enough such that it would be best for the work to be revised, more experiments to be run, and submitted to a future conference.  In general I can be very supportive of papers that have a strong empirical simulated result, even if there is no theory or real-world result.  Usually though, one would expect a paper which focuses on empirical results to provide an accurate interpretation and framing of those results.   In my opinion, the scope of revisions and open questions (see my list in previous section) is large enough such that it seems best to revise and submit to a future conference.

*Update after rebuttal:*
I think the paper is more clear now.  I will assume the authors will be willing to make the two edits below, which I think are required for truthfulness.

#1, Line 46, key moment of intro:
- current: "Our main contribution is a Surprisingly Simple Self-Supervised offline RL (S4RL) algorithm that combines studying data augmentations with a simple Q-learning method to significantly improve the performance of offline RL algorithm. "
- critiques of the current statement:
    - I don't think they contributed anything that can be called an algorithm? As I pointed out in my initial review, Equation 4 is directly from DrQ.
    - "Combining data augmentations with a simple Q-learning method" somewhat implies that they have a new Q-learning method, which they don't.
- suggested instead: "Our main contribution is in showing the effectiveness that simple data augmentations can have in improving existing offline RL methods."

#2, Line 88, key discussion of related work:

- current: "The closest work to S4RL include DrQ [14] and RAD [15] in that we utilize their method for Q-learning over augmentations, but in contrast we focus specifically on offline RL for robotic tasks and propose augmentations from states."
- critical error of the current statement:
    - RAD also proposed augmentations on states.
- suggested instead: ""The closest works to S4RL include DrQ [14] and RAD [15] in that we utilize their method for Q-learning over augmentations for online RL, but in contrast we focus specifically on offline RL for robotic tasks, and <insert some statement about the thoroughness of results>"

In contrast I think these statements at other key moments are more truthful and clear:

- Abstract: "In this paper, we study the effectiveness of performing data augmentations on the state space, and study 7 different augmentation schemes and how they behave with existing offline RL algorithms."
- Conclusion: "In this paper, we present S4RL: a Surprisingly Simple Self-Supervised offline RL method that uses data augmentations to improve the function approximation for Q-learning algorithms in offline RL."
    - nit: they haven't actually shown that they improve function approximation. They've just shown that the methods work better. But I consider this to be a minor grievance compared to #1 and #2 above.

If they could correct those statements I would be supportive of having the paper at the conference.

---

> ### Author Response · Authors · 2021-08-24
> **Thank you for your time! 1/2**
>
> Thank you for your time to review our draft! We address each point individually.
>
> > Novelty claim considering RAD and DrQ
>
>
> One of the major points of the paper was to empirically study the role of self-supervision and data augmentation techniques especially for state based augmentations methods in offline RL.
> While the methods above study structured perturbations in image spaces (for ex: random crops of the image), we study these in state based models.
> And while the initial goal was the offline rl setting, we has to our surprise found that the same methods work equally well in online settings as well.
> We additionally provide results on online RL, using a base SAC agent trained on 3 OpenAI Gym environments @ 1M timesteps where we report the episodic rewards(the same as the ones present in the D4RL tasks). We continue to see significant improvements in online RL as well :
>
> |                | SAC  | SAC+S4RL-N | SAC+S4RL-Adv |
> |----------------|------|------------|--------------|
> | HalfCheetah-v2 | 9302 | **11039**      | 10483        |
> | Hopper-v2      | 3341 | **3891**       | 3614         |
> | Walker2d-v2    | 3620 | **4367**       | 4007         |
>
>
> **The vital lesson for the community from this study is not just the novelty of yet another rl algorithmic, but a thorough analysis of data augmentations methods in state-spaces, and the unreasonable effectiveness of simple augmentations schemas in value function approximation**.
> We have provided extensive experiments using over 35 different data distributions from 16 different environments for continuous control tasks with 3 strong baselines in a variety of settings for continuous control tasks in a variety of settings: sparse & dense reward, easy vs hard control envs, robotics envs across a set of manipulation and locomotion tasks.
>
>
> > Is overfitting really a problem in offline RL? Would augmentation help overfitting?
>
>
> Overfitting is indeed a significant problem in offline RL. This has previously been explored in a blog [[1]](https://bair.berkeley.edu/blog/2020/12/07/offline/) that summarizes two recent papers [[2]( https://arxiv.org/abs/2006.04779), [3](https://arxiv.org/abs/2010.14500)] from the same authors (see second last paragraph of the blog).
>
> The blog explicitly states that “In a number of cases, this “overfitting” phenomenon gives rise to poorly-conditioned neural networks”. Augmentations enable better function approximation using NNs, in this case for Q-networks by ensuring that multiple augmentations of the same scene are mapped to the same latent space.
>
>
> > Why does dropout not work well since dropout is supposed to help with overfitting?
>
>
> Dropout has been empirically shown to be useful for computer vision tasks, especially with Bayesian networks. The behaviour of dropout in RL for robotics is largely unexplored, and it is unclear what may be the reason for this.
>
> However, that said there is some work attempting equivalence of data augmentation and dropouts (link), data augmentations are found to be empirically more efficient mechanisms rather than dropouts in convolutional deep learning architectures (Link, link2). Hence we pursue data augmentation as the primary avenue in this study.
>
> Nonetheless, the role of dropout in offline RL is orthogonal to this effort and likely a valuable question for future work. It is possible that a different form of MC-Dropout may be able to learn improved value functions.
>
> > Problems with “Self-supervision” when the method only uses data augmentations
>
>
> Thank you for pointing this out. From our perspective, data augmentation is the simplest form of self-supervision through which the models are trained using the average of different data augmentations.
>
> If you feel that the name is not representative enough, we would be open to suggestions and happy to modify for more accurate representation of the contribution.

---

> > ### Author Response · Authors · 2021-08-24
> > **Thank you for your time 2/2**
> >
> > > Small perturbations may lead to drastically different Q-values
> >
> >
> > While indeed this is counterintuitive, our results have suggested otherwise
> > Intuitively, one might argue that it is possible for small perturbations to lead to drastically different “true” Q-values, but through our experiments, we show how even in sparse reward settings (see Adroit and AntMaze experiments in Table 2), we are able to significantly outperform the different baselines. Usually that may only happen in very low state, action dimensions such a pendulum with very sharp value gradients.
> >
> >
> > A possible explanation for this effect could be the fact that perturbed states add support to the function approximator, thereby improving the lipschitz continuity of the fitted output, hence a better fit, especially in regions of state space with sparse coverage, which is often the case in large continuous state-action spaces in reinforcement learning.
> >
> >
> > The important thing to consider is since the input space is continuous, it is important for the Q-functions to learn values that are smooth locally, and “consistent” across augmentations.
> >
> >
> > Another important consideration is that in RL, accurate Q-function approximation everywhere is not necessary for large state-action spaces, as long as the mode predicts the right action. A near-optimal policy is achieved much sooner than the value function converges to true optimal in model-free RL. Hence while this effect may appear, it doesnt affect the agent performance adversely in most practical cases.
> >
> >
> > > The agent does not actually do “local exploration” as written
> >
> >
> > We agree that we use the term “exploration” a bit loosely, to mean that the agent is trained on data that is not directly from the dataset.
> > Instead it creates new data-points, not all of which would be physically realizable. However even in those cases they add to the support of the value function approximator improving the schitz continuity of the fitted output.
> > We will fixed this in the updated manuscript in line 122.
> >
> > > Is the noise from real world robots self-correcting to self-supervision?
> >
> >
> > It would be a conjecture to state that the noise from real robots is self-correcting.
> > We have added results in measurement model of the robot.
> > During testing, we randomly perturb the states by using an additive Gaussian noise of the form N(0, 0.001).
> > We report results on the same Metaworld tasks as in Figure 4 (left) and Table 3. In the results we see that we are able to significantly improve the generalization ability of the learned agents using S4RL.:
> >
> > |            | CQL | CQL+S4RL-N | CQL+S4RL-Adv |
> > |------------|-----|------------|--------------|
> > | push       | 49% | 69%        | **73%**      |
> > | door-close | 38% | 65%        | **67%**      |
> > | sweep-into | 21% | 50%        | **59%**      |
> > | reach-wall | 15% | 19%        | **28%**      |
> > | reach      | 10% | 13%        | **22%**      |
> >
> > We observe that the best performing agent in challenging robotic tasks (see Fig 4 and Table 3) is the agent trained with adversarial augmentations.
> > It is unlikely that any kind of real world noise would work in such a manner, which further shows the importance of the work.
> >
> > > Adding citations to relevant imitation learning literature
> >
> >
> > Thank you for the suggestions, we have now added all of the suggested references in the updated manuscript. A detailed discussion is also added in section 2..

---

> > ### Comment · Reviewer_oh9V · 2021-08-25
> > **Response on overfitting**
> >
> > You responded to this question: "Is overfitting really a problem in offline RL? Would augmentation help overfitting?"  but this was not the question I asked.
> >
> > The question instead is to respond to my comment "A central claim of the paper is that the presented method reduces overfitting, but there is insufficient data to support this claim."  Is there something you'd like to say in response to this?
> >
> > Thanks!

---

> > > ### Author Response · Authors · 2021-08-26
> > > **Experiments regarding overfitting**
> > >
> > > Thank you for the clarification.
> > >
> > > To test for overfitting, we performed additional experiments varying the network size. This is similar to the experiments done by [DrQ](https://arxiv.org/abs2004.13649)  (see Figure 1b and Section 3.1) published in ICLR 2021 as a spotlight, where they show that as the network sizes increase, the models tend to overfit more. We change the number of hidden units in each layer of the MLPs used to parameterize the models, and show their effect on training on the 3 hardest MetaWorld tasks. We change the MLP hidden units from 256 to 1024 and 4096 for both CQL and CQL+S4RL-Adv and report the task completion %.
> > >
> > >
> > > |            	| CQL - 256 (baseline) 	| CQL+S4RL-Adv - 256 (baseline) 	| CQL - 1024 	| CQL+S4RL-Adv - 1024 	| CQL - 4096 	| CQL+S4RL-Adv - 4096 	|
> > > |------------	|----------------------	|-------------------------------	|------------	|---------------------	|------------	|---------------------	|
> > > | push       	| 12%                  	| **35%**                       	| 11%        	| **33%**             	| 4%         	| **33%**             	|
> > > | door-close 	| 21%                  	| **39%**                       	| 19%        	| **37%**             	| 12%        	| **36%**             	|
> > > | sweep-into 	| 32%                  	| **79%**                       	| 27%        	| **74%**             	| 11%        	| **75%**             	|
> > >
> > >
> > > We can clearly see that as the number of hidden units is increased from 256 to 1024 and 1024 to 4096, we see a significant decrease in performance for the baseline CQL agent, while we only see a relatively small drop in performance for the CQL+S4RL-Adv agent showing how augmentations significantly decrease the overfitting of the model.

---

> > > > ### Comment · Reviewer_oh9V · 2021-08-26
> > > > **Still not evidence of overfitting, but some suggestions for you**
> > > >
> > > > Hi there, these experiments still do not support the claim that the method "prevents overfitting."  Also, a paper being published (and that's a nice paper) does not mean that every statement in it is 100% correct.
> > > >
> > > > Overfitting (https://en.wikipedia.org/wiki/Overfitting) means a specific thing, and it doesn't mean "performed badly in closed loop evaluations."  You can't analyze overfitting by only looking at success rates of closed loop evaluations.  Other papers might have done that, but it doesn't mean it's a correct analysis.
> > > >
> > > > One suggestion, Option A, for you would be to just remove the use of "overfitting" in your paper.   I.e., the solution might be to not add new experiments that do support the overfitting claim, but instead just to remove the overfitting claim.  You could instead just say "we don't know why this works better, but it seems to work better".
> > > >
> > > > A different option, Option B, would be to design some experiments that analyze overfitting.  For example you could set up an MDP for which you can compute the optimal Q function through value iteration, and then analyze whether your noise methods actually affect the validation loss of fitting the value function.
> > > >
> > > > Why not just do Option A?

---

> > > > > ### Author Response · Authors · 2021-08-27
> > > > > **Thank you for your response!**
> > > > >
> > > > > Thanks indeed for the suggestion of the overfitting experiment. However, we believe the suggested experiment by using a small MDP for which optimal value function is brute force computable is not going to work either. There are few, if any, non-tabular MDPs of meaningful scale where optimal value function can indeed be computed. In most of those cases, other simpler value estimation methods would work equally well as is the case with RL, many algorithms work well in simpler tabular settings but do not have the similar behavior in complex environments. Hence, a comparison in gridworld environment may not satisfy the scale of the experiments considered in the paper.
> > > > >
> > > > > Since the exact understanding of this overfitting remains an open question, particularly in non-tabular continuous state-action spaces, we agree with the reviewer and will accordingly update the draft. This would be the Option A, as suggested by the reviewer.
> > > > >
> > > > > We have appropriately changed the text to further reflect this in the abstract and lines 121-122, 162-163 and 223, where we have removed mentioning overfitting entirely as a driver for S4RL, to not mislead readers.

---

> > ### Comment · Reviewer_oh9V · 2021-08-25
> > **Response on state-based prior work**
> >
> > Hi there, I also made a comment on this on the Meta Review, but prior work did do state-based experiments, see Lashkin et al. "Reinforcement Learning with Augmented Data", https://arxiv.org/pdf/2004.14990.pdf, end of Section 4 and Section 5.4.  Given this, it seems like your novelty claim statements are not accurate.  Not sure if you'd like to say anything in response.
> >
> > Thanks!

---

> > > ### Author Response · Authors · 2021-08-25
> > > **Apologies for the confusion!**
> > >
> > > We apologize for the confusion, we did not mean to say that we are the first to look into state-based augmentations for RL.
> > > In fact, in the original draft, **we already compared against the best augmentation technique from RAD** in Table 1 (augmentation name: "random amplitude scaling"), and appropriately cite it when introducing the augmentations in the augmentation list in Section 4.2.
> > >
> > > We believe that this study still broadly is beneficial to the offline RL and robotics communities since it highlights the importance of data augmentation techniques for Q learning, which are intuitive in hindsight but were yet to be mainstream.
> > >
> > > We show thorough experimentation over a set of **35 different data distributions from 16 different environments for continuous control tasks** which include locomotion (Open AI Gym), manipulation (kitchen, adroit, MetaWorld, RoboSuite), and hierarchical planning (ant-maze) environments.
> > >
> > > ~~We have also started some experiments regarding your question on overfitting. We will post the experimental results soon.~~.
> > > **Overfitting results are now included below**: [Click link](https://openreview.net/forum?id=8xC5NNej-l_&noteId=t1iSyjXz70R)

---

### Official Review · Reviewer_bi2U · 2021-07-25

**Originality:** Excellent
**Technical Quality:** Good
**Clarity Of Presentation:** Very Good
**Impact:** 4

**Recommendation:**

Strong Accept: I recommend accepting the paper and will argue for my recommendation even if other reviewers hold a different opinion.

**Summary:**

This paper takes inspiration from recent improvements on image-base RL through augmentations and proposes to apply augmentations to state input as a form of self-supervision. The paper analyzes a range of different augmentations for offline RL by combining them with SAC (a state of the art RL method).

**Issues:**

L41,43,...: “proprioceptive observations” is not really true for the inputs used in all experiments as some of them will also provide the poses of objects to be manipulated (see Fig. 7).

L207: “we compare the algorithm to other self-supervision techniques that have been proposed for pixel-based RL”. This statement confuses me as it suggests that the proposed method works from pixel input where I had assumed that it is working from state input. Now, if the proposed method is actually working from pixel input (maybe applying an encoder to produce a low-dim. state and the augmenting that as described), this needs to be explained better. And if this is not the case and the proposed method actually works on state input, then comparing to pixel-based baselines should be done very very carefully as this is generally not a fair comparison.

Table 2: Here the question is now if “Normal” refers to learning from state or from pixels. And then +CURL/VAE or +S4RL cannot both be extensions on top of “normal” if one works from pixels and one works from state input. In this case, we need two “normal” versions and it should be made very clear in the table which results are based on pixel or state input (which is currently not indicated at all).

Table 1 & 2: In both tables, I have trouble finding definitions for the meaning of the different suffixes to the task names, e.g. “-medium”, “-medium-expert”, “-unmaze”, “-unmaze-diverse”, “medium-play”, “-human”, “cloned”. I would suggest to clearly explain all of these in the captions of these tables and potentially to split up the task name column into “task” and “data generation”.

Right now, the method is only evaluated in an offline setting and that is understandable if the authors are interested in that setting. But it would make this paper much stronger if the interactions of the online / offline setting and the advantage of using the proposed augmentations were evaluated. This way, we could test whether S4RL is especially suited for the offline setting as suggested e.g in L121-124.

**Reviewer Expertise:**

Good: General knowledge of the area

**Strengths And Weaknesses:**

The paper is clearly written, presents an idea that makes a lot of sense and appears simple in hindsight (I mean this as a big plus). The creative and original contributions are the augmentations considered here (Section 4.2).

The weaknesses of the paper are a few missing clarifications (see below) especially regarding the experiments and the limitation of the experiments to the offline setting – it would be interesting to see if the method produces similar improvements in online RL, or if it particularly fits the challenges of offline RL as suggested (but never shown) in the paper.

**Summary Of Recommendation:**

This paper appears correct and contains results that should be significant to the robot learning community. The paper is generally very well written and structured. If the issues regarding clarity and the experimental section are resolved, this can make a good CoRL paper.

EDIT: Issues addressed in rebuttal.

---

> ### Author Response · Authors · 2021-08-24
> **Thank you for your time!**
>
> Thank you for your time to review our draft! We address each point individually.
>
> > Pixel based input or state-based input… State based or pixel-based in table 2?
>
> All agents are trained with only state-based inputs in all experiments. We will highlight this and clarify in the final version of the paper. All baselines and S4RL agents are trained from states.
>
> > Describing the D4RL the data collecting mechanism
> That is a great suggestion! The data collecting mechanism is described in detail for the MetaWorld and RoboSuite experiments in App. F.
>
> We will add an additional section detailing the data collection process in the appendix in the final version. More details can be read in the original D4RL whitepaper [[1]](https://arxiv.org/abs/2004.07219).
>
> > Experiments with online RL?
>
>
> Thank you for the suggestion - we have added additional experiments with online RL using a base SAC agent trained on 3 OpenAI Gym environments @ 1M timesteps where we report the episodic rewards(the same as the ones present in the D4RL tasks). We continue to see significant improvements in online RL as well:
>
> |                | SAC  | SAC+S4RL-N | SAC+S4RL-Adv |
> |----------------|------|------------|--------------|
> | HalfCheetah-v2 | 9302 | **11039**      | 10483        |
> | Hopper-v2      | 3341 | **3891**       | 3614         |
> | Walker2d-v2    | 3620 | **4367**       | 4007         |

---

> > ### Comment · Reviewer_bi2U · 2021-08-27
> > **Review update**
> >
> > Thank you for the clarification and the additional results. As this addresses my main concern, I'll update my rating accordingly.

---

> > > ### Author Response · Authors · 2021-08-27
> > > **Thank you for your time!**
> > >
> > > Thank you for your time to check the rebuttal and for updating your score!

---

### Official Review · Reviewer_vBpT · 2021-07-26

**Originality:** Good
**Technical Quality:** Good
**Clarity Of Presentation:** Good
**Impact:** 3

**Recommendation:**

Weak Accept: I recommend accepting the paper, but will not argue for my recommendation if the majority of other reviewers have a different opinion.

**Summary:**

This paper aims to tackle the problem of offline reinforcement learning and presents an extension to existing methods via data augmentations. The method, S4RL: a Surprisingly Simple Self-Supervised offline RL method, assumes that the reward function is locally smooth and employs a range of data augmentation mechanisms to expand the offline dataset. The authors hope to improve the generalization ability of the RL agent in regions not well-supported by the original offline data.

The authors compare and benchmark the effectiveness of seven different data augmentation strategies from states in offline RL and then use the best-performing ones to compare against different self-supervised representation learning algorithms that have been proposed for pixel-based online RL.

**Issues:**

Please see the weakness section for detailed descriptions of the issues.

**Reviewer Expertise:**

Very good: Comprehensive knowledge of the area

**Strengths And Weaknesses:**

=== Strength

The proposed method, S4RL, is simple and easy to implement, making it compatible with many offline reinforcement learning algorithms that require Q-learning.

The authors show that the augmentation techniques can improve the performance over existing baselines.


=== Weakness

This paper is not particularly novel. Augmentation is a widely-used technique in a range of applications. It is not surprising that applying data augmentation to offline RL can improve the performance in some scenarios. It would improve the significance of this paper if the authors can show some observations that are specific to offline RL. For example, what are some of offline RL's properties that make it more suitable for some data augmentation techniques and not the others?

I also have concerns regarding the applicability of the proposed method. The authors make an assumption that the reward has to be smooth. However, many real-world tasks can involve discontinuous events like contacts. Considering grasping or pushing an object, the reward can be sharply different when the end effector is off by just a little bit. In cases where the end effector is close to the table, data augmentation could also lead to physically-infeasible states and incorrectly assign high value to these states. The authors may consider providing more theoretical justifications on where and when we should include the augmentation and to which extent does it help.

I also have questions regarding the comparison with the baselines. CURL [1] and CQL [2] were designed or at least contain experimental results where the inputs to the agent are visual observations. What are the inputs to S4RL in each experimenting environment? Are they images or state information? The authors should include more detailed descriptions of the testing environments for the readers to better understand the context of the reported performance scores.

Related to my previous point, if the inputs to S4RL contain high-dimensional visual observations, why do the authors only apply augmentations to the state? What about the wide range of augmentation techniques that can be applied to images? If the inputs to S4RL are low-dimensional states, comparing it with techniques like CURL and CQL does not seem particularly convincing to me, as they are designed and have shown results in more challenging scenarios than this paper.

Typo: the symbol for the dataset in equation 3 is not consistent with the text description.

[1] Srinivas et al., CURL: Contrastive Unsupervised Representations for Reinforcement Learning
[2] Kumar et al., Conservative Q-Learning for Offline Reinforcement Learning

========

Post-rebuttal

I have read the reviews from other reviewers, the rebuttal from the authors, and the discussions. I appreciate the authors' efforts in providing detailed responses and the additional experiments, which have addressed my major concerns. The detailed comments from Reviewer oh9V and the revision from the authors also helped make the statements in the paper more precise at describing the contributions and related concepts.

I'm now supportive of this paper to present at the conference and have updated my score from Weak Reject to Weak Accept.

**Summary Of Recommendation:**

I have concerns regarding the novelty and the applicability of the underlying assumptions of this paper. I also find the comparison with the baselines not particularly fair, as the baselines are designed to process high-dimensional visual observations, and this paper, S4RL, only seems to apply augmentation to the low-dimensional states.

---

> ### Author Response · Authors · 2021-08-24
> **Thank you for your time!**
>
> Thank you for your time to review our draft! We address each point individually.
>
> > Not particularly novel… not surprising results… what are the properties of offline RL that makes it suitable for augmentation techniques?
>
>
> **The vital lesson for the community from this study is not just the novelty of yet another rl algorithmic, but a thorough analysis of data augmentations methods in state-spaces, and the unreasonable effectiveness of simple augmentations schemas in value function approximation**.
> We have provided extensive experiments using over 35 different data distributions from 16 different environments for continuous control tasks with 3 strong baselines in a variety of settings: sparse & dense reward, easy vs hard control envs, robotics envs across a set of manipulation and locomotion tasks.
>
> One of the major motivation and outcome of the paper is **a thorough analysis of the role of self-supervision and data augmentation techniques.**
> Similar methods have become mainstay in computer vision and even in online pixel-based RL to an extent.
> The valuable outcome to share with the community is that a battery of these methods do not work despite their complexity, while simpler ideas work **surprisingly robustly and consistently across many experimental domains.**
>
>
> Our framework is useful for Offline RL as demonstrated, but also in Online RL, as supported by additional experiments using a base SAC agent. We  train on 3 OpenAI Gym environments @ 1M timesteps where we report the episodic rewards (the same as the ones present in the D4RL tasks).
>
> We continue to see significant improvements in **online RL** as well. We will add these results in the experimental section in the final draft:
>
> |                | SAC  | SAC+S4RL-N | SAC+S4RL-Adv |
> |----------------|------|------------|--------------|
> | HalfCheetah-v2 | 9302 | **11039**      | 10483        |
> | Hopper-v2      | 3341 | **3891**       | 3614         |
> | Walker2d-v2    | 3620 | **4367**       | 4007         |
>
> > The augmentations may lead to physically unrealisable states… some tasks have discontinuous reward distributions where augmentations may hurt
>
>
> Thanks a lot and this is an indeed reasonable observation.
> Counterintuitively, we have observed that adding small perturbations to observed states results in regularization of the value/Q function approximations, despite some of them not being realizable.
> A similar effect is also observed in image space perturbations such as  “color jitters” and “random crops”, which also result in unrealizable observations, yet improve performance nonetheless.
> A possible explanation for this effect could be the fact that perturbed states add support to the function approximator, thereby improving the lipschitz continuity of the fitted output, hence a better fit, especially in regions of state space with sparse coverage, which is often the case in large continuous state-action spaces in reinforcement learning.
> We evaluate this argument in environments with non-smooth reward functions (as mentioned by the reviewer).
> For instance, ant-maze (Table 2) has a reward function wherein the ant agent gets a +1 reward when it gets to the “goal” position in a maze, receiving a reward of 0 otherwise (highly sparse rewards).
> Another environment with highly sparse rewards are the “Adroit” environments (Table 2) where Shadow Hand agent receives *a positive reward only when the pose of the object the hand is manipulating is oriented correctly*
> We observe that S4RL augmentations are very useful in both of these environments (as listed in Table 2), even with discrete reward functions!
> We note that the fitted Q-network is still continuous, hence the augmentations help in improving function approximation.
>
>
> > State or image input to the agents?
>
> All the agents trained are trained using **state-information only**.
> We will clarify and highlight this in the final version in Section 5 (Experiments).
>
>
> > If state info is the input, then do the baselines make sense?
>
> Even though all the agents are trained only with state-information (including CQL+CURL and CQL+VAE) the baselines are still reasonable, and were tuned specifically for the tasks. Furthermore, it is clear to see that these methods are effective even when working from states as CQL+VAE and CQL+CURL almost always improve the baseline CQL agent.
> We will add the chosen hyperparameters in the final version for full clarity.

---

> > ### Author Response · Authors · 2021-08-28
> > **Update on the rebuttal**
> >
> > We have attempted to answer with a point-wise rebuttal.
> > We were wondering if you had a chance to look at the rebuttal and if there remain any questions/clarifications that we can answer.
> > We hope that your constructive feedback with lead to a technically stronger contribution.

---

### Meta-Review · Area_Chair_fQF7 · 2021-08-10

**Recommendation:** Accept (Poster)
**Confidence:** 4

**Metareview:**

After an extensive discussion, the reviewers changed their scores to indicate that the paper should be published at CoRL. I would like to encourage the authors to carefully analyze the reviews and address the reviewers' points (especially regarding some of the wording pointed out by Reviewer oh9V) in the final version of the paper.

---

> ### Author Response · Authors · 2021-08-24
> **Thank you for your time to review the draft!**
>
> We thank the reviews for their time to review our draft. The reviewers found the paper to be well written and “make a lot of sense” [R-bi2U, R-TxK8], simple and easy to implement for offline rl using Q learning [R-vBpT], while achieving state-of-art numbers on the D4RL benchmark [R-oh9V], and addresses an important direction in the field [R-TxK8].
>
> We have addressed some common concerns here:
>
>
> > inaccurate statements about the novelty of the contribution
>
> We emphasize that the vital lesson for the community from this study is not just the novelty of yet another rl algorithm, but a thorough analysis of data augmentation methods in state-spaces, and the unreasonable effectiveness of augmentation schemas in robust value function approximation in continuous control domains which are relevant to robotics
>
> > state-based data augmentation can lead to infeasible and unstable states
>
> We address a common question that each of the reviewers had in that all the experiments are done with state-based input. Previously, such work has been proposed for pixel-based RL which highlights which augmentation techniques are effective [[1]](https://arxiv.org/abs/2004.14990), but such a study is currently lacking for state-based inputs. We try to provide such a study in the context of offline and online RL (results below).
> A possible explanation for this effect could be the fact that perturbed states add support to the function approximator, thereby improving the lipschitz continuity of the fitted output, hence a better fit, especially in regions of state space with sparse coverage, which is often the case in large continuous state-action spaces in reinforcement learning.
>
>
> > clarifications on the experiment settings regarding experiments exclusively on offline rl
>
> We have  provided extensive experiments in the **offline RL** setting, where we experiment with **35 different data distributions from 16 different environments** for continuous control tasks.
>
> Most of the presented results are in offline settings, but we have additionally ran experiments on online rl (also in Appendix B), where we continue to observe improvement over the baseline SAC agent.
>
> Given the time constraints, we have also added limited results on online RL using a base SAC agent. We train on 3 OpenAI Gym environments @ 1M timesteps where we report the episodic rewards (the same as the ones present in the D4RL tasks).
>
> We continue to see significant improvements in **online RL** as well. We will add these results as well as expanded online RL numbers in the experimental section in the final draft:
>
> |                | SAC  | SAC+S4RL-N | SAC+S4RL-Adv |
> |----------------|------|------------|--------------|
> | HalfCheetah-v2 | 9302 | **11039**      | 10483        |
> | Hopper-v2      | 3341 | **3891**       | 3614         |
> | Walker2d-v2    | 3620 | **4367**       | 4007         |
>
>
>
>
>  > is overfitting really an issue for offline RL?
>
> Overfitting is indeed a significant problem in offline RL. This has previously been explored by Kumar et al. ( [[blog]](https://bair.berkeley.edu/blog/2020/12/07/offline/) based on [[2](https://arxiv.org/abs/2006.04779), [3](https://arxiv.org/abs/2010.14500)]).
> The authors [1,2,3] state that “In a number of cases, this “overfitting” phenomenon gives rise to poorly-conditioned neural networks”. In view of this empirical observation, we study mechanisms that utilize augmentations, instead of dropout as suggested by Reviewer oh9V . Augmentations enable better function approximation using NNs, in this case for Q-networks by ensuring that multiple augmentations of the same scene are mapped to the same latent space.
>
>
> > questionable classification of the method as a self-supervised approach
>
> From our perspective, data augmentation is the simplest form of self-supervision through which the models are trained using the average of different data augmentations.
> If reviewers feel that the name is not representative enough, we would be open to suggestions and happy to modify for more accurate representation of the contribution.

---

> > ### Comment · Area_Chair_fQF7 · 2021-08-24
> > **Thank you for the response!**
> >
> > Thank you for the response! This is very helpful in determining the final decision for the paper.
> >
> > I will wait for the reviewers to reply to the points you raised before I provide my own assessment.

---

> > ### Comment · Reviewer_oh9V · 2021-08-24
> > **Response on state-based inputs.**
> >
> > Sorry authors, but prior work did do state-based experiments, see Lashkin et al. "Reinforcement Learning with Augmented Data", https://arxiv.org/pdf/2004.14990.pdf, end of Section 4 and Section 5.4. Given this, it seems like your novelty claim statements are not accurate. Not sure if you'd like to say anything in response.
> >
> > Thanks!

---

> > > ### Author Response · Authors · 2021-08-25
> > > **Apologies for the confusion!**
> > >
> > > We apologize for the confusion, we did not mean to say that we are the first to look into state-based augmentations for RL.
> > > In fact, in the original draft, **we already compared against the best augmentation technique from RAD** in Table 1 (augmentation name: "random amplitude scaling"), and appropriately cite it when introducing the augmentations in the augmentation list in Section 4.2.
> > >
> > > We believe that this study still broadly is beneficial to the offline RL and robotics communities since it highlights the importance of data augmentation techniques for Q learning, which are intuitive in hindsight but were yet to be mainstream.
> > >
> > > We show thorough experimentation over a set of **35 different data distributions from 16 different environments for continuous control tasks** which include locomotion (Open AI Gym), manipulation (kitchen, adroit, MetaWorld, RoboSuite), and hierarchical planning (ant-maze) environments.

---

### Decision · Program_Chairs · 2021-09-13

**Decision:**

Accept (Poster)

**Comment:**

After an extensive discussion, the reviewers changed their scores to indicate that the paper should be published at CoRL. I would like to encourage the authors to carefully analyze the reviews and address the reviewers' points (especially regarding some of the wording pointed out by Reviewer oh9V) in the final version of the paper.